# Monitoring the Functionality and Stress Response of Yeast Cells Using Flow Cytometry

**DOI:** 10.3390/microorganisms8040619

**Published:** 2020-04-24

**Authors:** Stephan Sommer

**Affiliations:** Viticulture and Enology Research Center, California State University, 2360 E. Barstow Ave, Fresno, CA 93740, USA; ssommer@csufresno.edu; Tel.: +1-559-278-0277

**Keywords:** flow cytometry, *Saccharomyces*, physiology, fermentation, process control

## Abstract

Throughout fermentation, yeast faces continuously changing medium conditions and reacts by adapting its metabolism. The adaptation is a critical process and is dependent on the accurate functioning of the cell. A stable membrane potential, which is, among other roles, responsible for protecting the yeast from low pH, is an important attribute for evaluating functionality. Other factors are storage products such as glycogen, trehalose, and neutral lipids, as well as mitochondrial activity and the integrity of the DNA. These parameters can be complemented by the analysis of viability, cell cycle, intracellular pH, and reactive oxygen species in the cell. The correlation of all these factors provides valuable information for evaluating the performance of a yeast population during fermentation. In order to demonstrate the analytical capabilities of flow cytometry, a *Saccharomyces cerevisiae* yeast strain was observed in a modified growth medium for 384 h (16 days). The results confirm observations made with other methods and reports from the literature. However, with flow cytometry, it is possible to gain deeper insight into stress response and adaptation behavior of yeast at a cellular level. The causality from the formation of oxygen-radicals to cell death, for example, can be shown, as well as the dependency of the intracellular pH on the stability of the membrane. The proposed bio-monitoring system has the potential to provide applicable information as a process control tool for wineries.

## 1. Introduction

During alcoholic fermentation, yeast faces a variety of stress factors that influence its metabolic performance and proliferation rate. Grape juice is commonly inoculated at about 18 °C. In the very beginning of white wine fermentation, the winemaker usually applies relatively low temperatures around 15 °C to slow down yeast metabolism and preserve volatile aroma compounds, as fermentation vessels heat up due to the heat energy derived from sugar conversion. *Saccharomyces cerevisiae* has a higher temperature optimum, which causes an initial compensative stress response in order to adapt to the must medium [1]. Low-level stress is believed to be beneficial for overall wine quality due to potential over-production of aroma compounds [2]; however, the acceptable level is strain dependent and cannot be generalized [3,4]. Too much pressure on the detoxification and stress response mechanisms of the cell can lead to stuck or sluggish fermentation [5,6] and can decrease wine quality, which can lead to economic losses for the producer.

High sugar levels in the beginning and increasing alcohol concentrations later in the course of fermentation confront *Saccharomyces* with more unfavorable conditions. High osmotic pressure and ethanol toxicity make the cell less resistant to other stress factors and can contribute to fermentation problems [4]. Low medium pH plays an important role in the stress response of yeast because cells are trying to maintain a stable neutral pH in the cell against the proton gradient in the medium [7]. A drop in intracellular pH indicates a malfunction of the proton pump due to membrane failure and therefore a decrease in metabolic activity [8]. The functionality of yeast depends on the ability of the cell to defend itself from unpleasant medium conditions. In particular, reactive oxygen species such as hydrogen peroxide (H_2_O_2_) or peroxyl radicals (COO^−^) can cause serious damage to the stability of the cell [1] and the integrity of the DNA [9]. *Saccharomyces* is reported to adapt to some stress factors rather quickly, while others lead to more severe stress responses [4].

Glycogen, comparable to starch in plants, is one of the most important storage carbon sources for yeast [10]. The level inside the cell can reach up to 30% of the dry mass [11]. Neutral lipids on the other hand are reported to be part of the defense mechanism against low temperatures, high alcohol, and low pH conditions [12]. Monitoring storage compounds in general provides valuable information about the adequate supply of nutrients in the medium, the level of stress, and therefore the living conditions of the yeast [13]. Analyzing all these factors with conventional laboratory methods can be expensive and time-consuming. Even fluorescence-based methods that are commonly performed using microscopy only analyze a limited number of cells at a given time.

The system of flow cytometry was originally developed in modern medicine. It is widely used to detect fluorescence signals of large numbers of cells, which makes it very useful to provide a deeper insight into yeast life and the metabolism of a larger population [14,15]. With the flexibility of wavelengths and detection systems, a large variety of parameters can be analyzed simultaneously with very little preparation time [16].

The objective of this work was to demonstrate the use of flow cytometry as a modern analytical tool for fermentation monitoring and to visualize how and when wine yeast responds to stress factors caused by the changing conditions of alcoholic fermentation.

## 2. Materials and Methods

### 2.1. Culture Preparation and Sampling

The methods used for these experiments are based on or were developed from the work of Hutter et al. 1978–2002 [11,17,18,19,20,21].

In the present test -series, the commercial *Saccharomyces cerevisiae* yeast SIHA 8 (Eaton Begerow Technologies GmbH, Langenlonsheim, Germany) was grown and monitored in a Yeast Extract Peptone Dextrose (YPD) broth growth medium (MilliporeSigma, Burlington, MA, USA) with added sucrose (total of 20 Brix) for 384 h (16 days) and analyzed for its viability and functionality parameters. Proliferation of the inoculum was induced by aeration and agitation at 20 °C for twelve hours and, to synchronize the metabolism and cell cycle, the inoculation culture was stored overnight at 4 °C. This treatment leads to cell cycle arrest in the G_1_ phase and allows the culture to develop synchronously. Fermentation was performed in 20 L carboys in duplicate.

In the beginning of the experiments, hourly sampling intervals were chosen to monitor the adaptation phase of the yeast. After the first 24 h, the yeast adapted its metabolism, and samples were taken only once a day. The necessary sampling volume depends on the current cell density and should be adapted to the stage of fermentation, ranging from 20 mL in the beginning when the cell count is low to 5 mL later during active fermentation. It is advisable to have at least 10^6^ cells per mL in a sample for analysis. After sampling, yeast cells were washed twice in phosphate-buffered saline at pH 7.2 (PBS) (VWR International, Radnor, PA, USA) and fixed in 70% ethanol for cell cycle and storage product analysis. The other protocols use fresh cells, so dyeing can be done right after sampling and washing.

### 2.2. Reagents and Chemicals

All chemicals and reagents were purchased as pure substances through VWR (VWR International, Radnor, PA, USA). Stock and working solutions were prepared as described in the following sections.

### 2.3. Cell Cycle Analysis

The cell cycle analyses were conducted according to Hutter and Eipel [19] with minor modifications by measuring the DNA content of the cell.

Fixed yeast cells were washed in PBS and re-suspended in 37 °C RNAse-solution (1 mg RNAse/mL PBS pH 7.2). The incubation time of one hour must be exactly followed. The reaction was stopped by adding 5 mL of ice-cold PBS. After another washing step in PBS, the pellet was re-suspended in 1 mL PBS, and 100 μL of propidium iodide solution (7 mg PI/100 mL TRIS-buffer pH 7.5) was added. The samples must be kept in the fridge for at least one hour prior to analysis but dyeing shows the best results when stored at 4 °C overnight.

### 2.4. Intracellular Glycogen

The specific staining of glycogen inside the cells was carried out with acriflavine according to Hutter [18] with minor modifications.

Schiff’s reagent: 0.5 g potassium bisulfite was added to 75 mL water. An acriflavine-solution (0.5 g acriflavine in 15 mL concentrated HCl) was added, and the flask was filled up to 100 mL with demineralized water. Then, 24 h later, the solution was filtered with 300 mg charcoal and filled in a brown glass bottle. The stock solution should be stored cool and protected from light.

Fixed yeast cells were washed twice in PBS. The pellet was re-suspended in 1 mL periodic acid (0.5%) and incubated for ten minutes. After washing twice with PBS, 1 mL of the diluted Schiff’s reagent (1:100 with PBS) was added. The incubation time of one hour should take place at room temperature in the dark. After the reaction time, the cells must be washed at least twice or until the PBS remains colorless after washing to reduce the background noise for analysis.

### 2.5. Viability

Viable yeast cells were washed twice in PBS to remove any medium residue. The pellet was re-suspended in 5 mL pre-warmed PBS at 37 °C. Then, 100 μL fluorescein diacetate solution (0.5 g FDA in 100 mL ice-cold acetone) was added. After incubating for five minutes in the dark at room temperature, 200 μL propidium iodide solution was added (7 mg PI/100 mL TRIS at pH 7.5).

Dead yeast cells show a red fluorescence signal; living cells glow bright green.

### 2.6. Intracellular Neutral Lipids

Fixed yeast cells were washed twice and re-suspended in 1 mL PBS. After that, 20 μL Nile Red solution (1 mg Nile Red per mL acetone) was added. The sample can be analyzed after exactly 30 min incubation time at room temperature in the dark.

### 2.7. Mitochondria Activity

Viable yeast cells were washed twice and re-suspended in 2 mL PBS. After adding 10 μL Rhodamine 123–solution (50 μM), the sample was incubated for five minutes in the dark at room temperature. The fluorescence should be evaluated under a microscope prior to flow analysis.

### 2.8. Intracellular Reactive Oxygen Species

Viable yeast cells were washed twice and re-suspended in 250 μL PBS. After the addition of 20 μL 2′,7′-dichloro-dihydro-fluorescein diacetate working solution (1 mM CM-H_2_DCFDA in water-free DMSO and diluted to 5 μM with PBS), the sample was incubated for 45 min at 37 °C. The pellet was washed with 3 mL PBS and again incubated in the dark at room temperature for about one hour.

The stock solution can be stored at −20 °C in light-protected bottles to prevent any oxygen contact. It is beneficial for the shelf-life of the solution to protect it with nitrogen.

### 2.9. Degree of DNA Damage

Viable yeast cells were washed twice and re-suspended in 2 mL PBS. After adding 40 μL acridine orange solution (40 μM), the sample was incubated for five minutes in the dark at room temperature.

### 2.10. Membrane Potential

Viable yeast cells were washed twice and re-suspended in 2 mL PBS. After adding 40 μL DiBAC4(3) working solution (30 μM bis-(1,3-dibutylbarbituric acid) trimethine oxonole), the sample was incubated for five minutes in the dark at room temperature. The fluorescence should be evaluated under a microscope prior to flow analysis.

### 2.11. Intracellular pH Value

Viable yeast cells were washed twice and re-suspended in 2 mL PBS. After adding 40 μL BFECF-AM solution (10 μM), the sample was incubated for exactly fifteen minutes in the dark at room temperature.

### 2.12. Data Analysis

The analyses were performed on a CyFlow SL Cytometer (Partec, Münster, Germany) at 488 nm. The sheath-fluid was demineralized water with 0.01% sodium azide to inhibit algal growth. Samples were filtered through a 50 μm CellTrics® filter (Partec, Münster, Germany) prior to analysis. Partec FlowMax software (Partec, Münster, Germany) was used for data analysis. Data handling and statistical analysis were performed using SigmaPlot 12.5 (Systat Software Inc., San Jose, CA, USA).

## 3. Results and Discussion

### 3.1. Interpretation of Flow Cytometry Data

When displaying data, it is important to consider the scaling of each axis and the overall resolution of the histogram. Figure 1 shows some examples of histograms produced by flow cytometry. Most attributes should be displayed on a logarithmic scale due to the necessity to fit a wide range of fluorescence intensities on a short scale. The numerical data extracted from peak maxima can only be interpreted relative to one another if they are produced with the same settings. Changing the photomultiplier settings in the instrument will automatically change the numeric peak value and render a comparison meaningless. Keeping the settings constant throughout an experimental series is therefore essential.

For absolute intensity values, in the case of cell cycle analysis, for example, it makes sense to use a linear scale. Haploid and diploid strains are much easier to differentiate on that scale as can be seen in Figure 1. If the cell density during testing exceeds 10^7^ cells per mL, histograms get broader due to cell clumping. Analyzing a more dilute cell suspension, however, is always a better approach.

Most parameters can be displayed in a two-dimensional histogram plot, whereas viability should be visualized in a special dot-plot with the two different fluorescent dyes on the axes. In the upper left corner of the plot where green fluorescence intensity is highest, only viable cells are counted. In the bottom-right part, the dead cells with red fluorescence are located. The other two sectors include damaged or dying cells, and debris. By setting specific gates for data collection, it is possible to differentiate between and quantify dead and living cells without counting background noise and nonspecifically dyed particles.

Side peaks and sub-populations at different stages of fermentation are usually an indicator of uneven sampling or uneven cell distribution in a fermentation vessel. These phenomena can depend on the dimensions of the tank and the physical mixing behavior [22] as well as the sampling strategy of the winemaker.

Cellular storage attributes such as glycogen depend on the nutrient situation in the medium and are therefore similar in most of the cells. Double peaks and sub-populations in storage carbohydrates can be an indicator of metabolically diverse cells, for example, when the sample includes dead and living cells in one trial. Figure 2 shows examples where two distinct peaks were visible in two separate experiments during the analysis of glycogen and trehalose in *S. cerevisiae* yeast cultures. Samples were purposefully taken from the sediment and the yeast in suspension to illustrate the effect of metabolic differences.

Trehalose, a disaccharide, changes more rapidly in the cell than glycogen. Differences between proliferating cells and starving yeast are also much more visible in a flow cytometry analysis. Glycogen is a long-term storage carbohydrate for *Saccharomyces* and plays an important role in cell recovery and survival under fermentation conditions [18], while trehalose is only used for short-term glucose storage.

Another critical attribute to evaluate the performance of a population is the cell cycle. Figure 3 shows the typical development of cell density during fermentation in combination with the cell cycle analysis of a haploid yeast strain. It is visually intuitive to differentiate the proliferating and fermenting cells by comparing the percentage in G_1_ and G_2_ phases, with the first peak representing cells in G_1_ and the second peak showing the percentage in G_2_. The area between the two peaks represents yeast cells that are actively doubling their DNA (S phase). Combined with the information of how many cells are viable at that specific moment, the system helps to determine the growing state of the fermentation culture.

### 3.2. Correlation of Physiological Yeast Data

Since the experiments were carried out in a small-scale format, the fermentation time of 16 days is relatively short. On the other hand, the cell density tends to be higher in these small-scale fermentations, which leads to higher fermentation efficiency [23] but also earlier contact inhibition and nutrient deficiency. Therefore, the stress response and adaptation behavior of yeast can be analyzed in a shorter period of time.

The results show confirmation of physiological data that are reported in the literature. The causality of the formation of oxygen radicals to a loss of viability, for example, can be shown with flow cytometry (Figure 4). Oxygen in the mitochondria of the cell is used to produce energy, which can lead to the formation of free radicals such as hydrogen peroxide (H_2_O_2_) or peroxyl radicals (COO^−^) [24]. These molecules can damage the DNA and therefore lead to cell death.

There are different pathways and biochemical processes leading from reactive oxygen species (ROS) to either apoptosis or necrosis. However, in the experiments presented here, there is no distinction between these two forms of cell death; only the degree of DNA damage and the final death rate were analyzed because in industrial fermentation, it is not of final interest which pathway lead to DNA fragmentation. Only the observation that there is a connection between reactive oxygen species and cell death is important at that point.

Since the most damaging oxygen species, hydrogen peroxide and peroxyl radicals, were included in the current method, it is possible to correlate the amount of oxygen in the cell with the degree of DNA damage and viability. After the initial drop in oxygen radicals in the first 48 h, the level increases and reaches a maximum after 120 h. Interestingly, the curve for mitochondria activity, reactive oxygen species, and DNA damage follow a very similar pattern. The reasons were previously explained by the relationship between oxygen-related energy production in the mitochondria and the release of ROS during that process [24]. However, the peak intensities that are visible around 120, 170, and 300 h do not immediately coincide with a direct loss of viability. One possible explanation is that the yeast culture was not actively growing at this late point in the process, and any damage to the reproductive mechanisms had only minor or no effects on the viability. Oxygen radicals in younger cultures or during proliferation can have detrimental effects on the onset and progression of fermentation by affecting the membrane integrity of yeast cells [16].

It is also possible to relate the membrane potential directly to the intracellular pH of yeast (Figure 5). Although the pH inside the cell is slightly delayed due to a change in membrane potential, the similar progression of both graphs is clearly visible. In the first 24 h, *Saccharomyces* raises the pH in the cell by stabilizing the membrane. In fermenting wine, the yeast medium has a pH of about 3.2 to 3.6, which is a minor problem when the H^+^-pump of the membrane is working, regulating the intracellular pH [25]. A decreasing membrane potential leads to a drop of the intracellular pH and therefore to the activation of autolysis enzymes such as proteases that degrade cell components leading to greater perforation of the cell [26].

The survival of the cell directly depends on the functionality of the membrane. With rising ethanol levels in the medium, the environmental conditions become rapidly unpleasant and some parts of the yeast population react by inducing cell death (Figure 4). Alcohol makes the membrane more fluid and therefore less resistant [27].

In these experiments, the analysis of the glycogen reserves shows a very typical behavior for wine fermentations (Figure 6). After a short lag-phase at the very beginning, yeast builds up glycogen to store glucose inside the cell. Yeast works as a single-cell organism isolated from others for most of its life and is therefore continuously competing for nutrients with all the other cells in the medium. Storing glucose and therefore withdrawing it from the medium has some advantage in the competition for nutrients.

In the following course of fermentation, *Saccharomyces* slowly consumes the glycogen. Figure 6 clearly indicates that the viability of yeast cells depends on the pool of storage carbohydrates since both graphs show a similar progression. In normal fermentation, it is very difficult to come to the point where no glycogen is left in the cell because the fermenting yeast will slow down its metabolism to preserve valuable energy sources, especially glycogen [28]. Due to this adaptation, *Saccharomyces* can survive for a very long time without any sugar in the medium. Low levels of glycogen at any point in the process indicate nutrient deficiency or enzyme malfunction [28] and can lead to stuck and sluggish fermentation.

Neutral lipids play an important role in the yeast’s defense mechanism during alcoholic fermentation [29]. Although lipids have a high energy density, they are usually not used as an energy resource but rather as a building block for cell membranes. Rapid increases in lipids in a fermenting culture are an indicator of stressful conditions and an active stress response of the cell [16]. Figure 6 combines the cytometric analysis of lipids in the cell membrane with glycogen and viability.

The concentration of neutral lipids increases during proliferation and in the first few hours of a fermentation during the adaptation phase. As soon as the yeast produces ethanol, the cellular lipids stay at a relatively high level due to the increasing solvent pressure of ethanol. With decreasing viability, the relative concentration of lipids in a culture can decrease because of cell lysis and the relative loss of membrane lipids. The sharp decrease followed by an increase of lipids in Figure 6 towards the end of the experiment could be explained by the selective pressure of ethanol at this point. While about half the population is already dead, the remaining cells are equipped with a lipid profile that would allow them to withstand the solvent effect of ethanol. Interestingly, the increase in lipid fluorescence intensity coincides with an increase in membrane potential (Figure 5), which supports the hypothesis that the sample contained mainly strong and well-adapted cells at this late fermentation stage.

Since research and method development in flow cytometry are continuously being expanded, there will be more parameters available in the future to assess the state and function of cells. If applied correctly, flow cytometry is flexible enough to analyze any activity and metabolite in the cell. Figure 7 provides an overview over the range of attributes that can currently be analyzed by flow cytometry.

## 4. Conclusions

*Saccharomyces cerevisiae* can be efficiently monitored by flow cytometry because of its cell size and great accessibility with respect to fluorescent dyes. Analyzing several different cell functions during fermentation allows the operator to establish correlations between cellular attributes and make predictions about the overall condition and stress response level of the culture. Cell cycle and viability, for example, provide information about the percentage of cells that are actively growing compared to fermenting, alive but inactive, or dead cells. In the present study, further correlations between intracellular macromolecules such as glycogen and neutral lipids and yeast viability as well as stress responses were shown. Other cellular functions such as mitochondrial metabolism can cause the formation of oxygen radicals that lead to a weakening of the culture and consequently, performance problems. These relationships are difficult to show with other techniques but can be included in a bio-monitoring system using just one instrument.

Flow cytometry has enormous potential as a process control tool in wineries because of the analytical flexibility and the wide range of organisms that can be monitored. Wineries can benefit from better insight into the yeast’s fermentation behavior and understand stuck and sluggish fermentations and the potential remedies that are available. Especially in challenging matrices such as late harvests and sparkling wine, a wider range of stress factors influence the fermenting culture and need to be addressed differently. Understanding the physiological response of the specific yeast strain might be the most important tool for successful fermentation.

The ability of flow cytometry to not only monitor fermentation performance of a statistically relevant portion of the yeast population but also identify potential fermentation problems at a cellular level before they affect the whole culture makes this technology a truly flexible quality control tool.

## Figures and Tables

**Figure 1 microorganisms-08-00619-f001:**
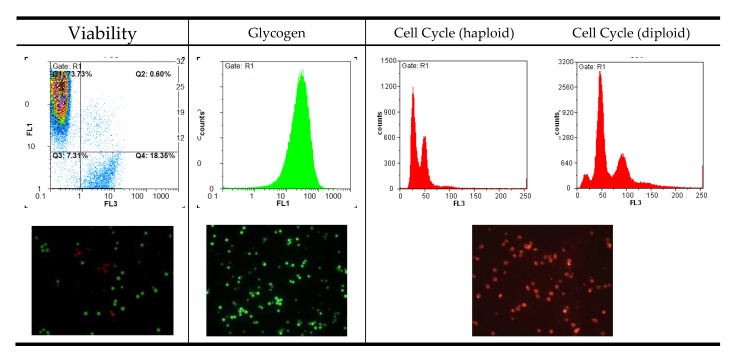
Example histograms and microscopic images to visualize the result output of flow cytometry (FL1: fluorescence channel green, FL3: fluorescence channel red).

**Figure 2 microorganisms-08-00619-f002:**
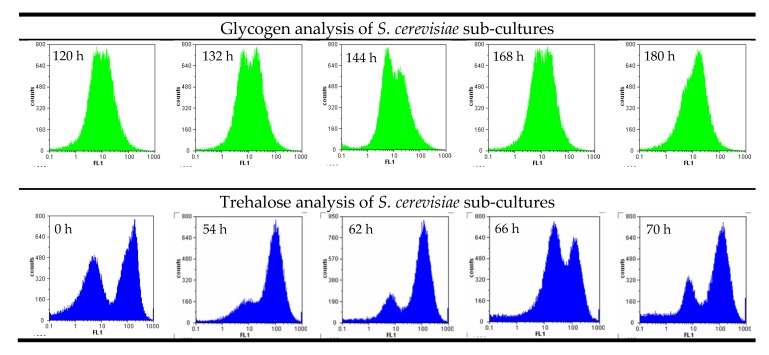
Examples of sub-cultures developed in yeast fermentations due to inhomogeneous conditions analyzed by flow cytometry.

**Figure 3 microorganisms-08-00619-f003:**
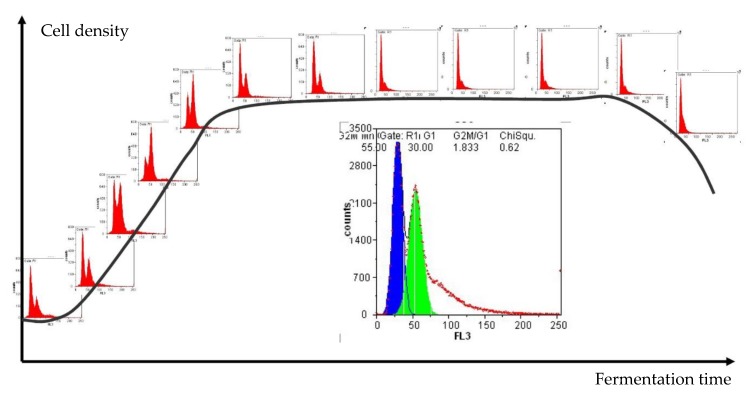
Yeast growth phases during fermentation analyzed by flow cytometry.

**Figure 4 microorganisms-08-00619-f004:**
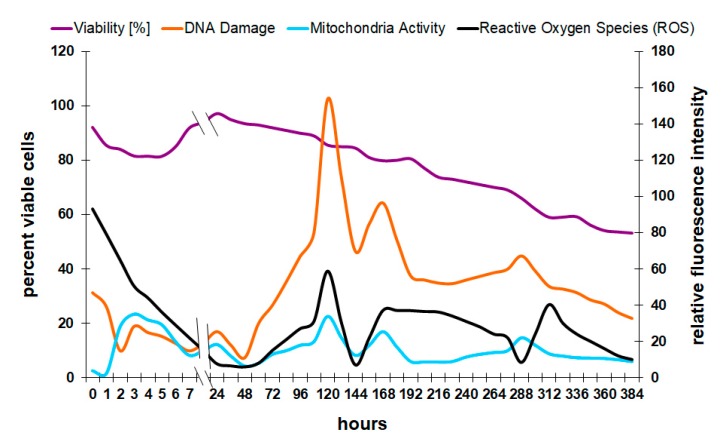
Yeast viability in connection with reactive oxygen species (ROS), DNA damage, and mitochondrial activity.

**Figure 5 microorganisms-08-00619-f005:**
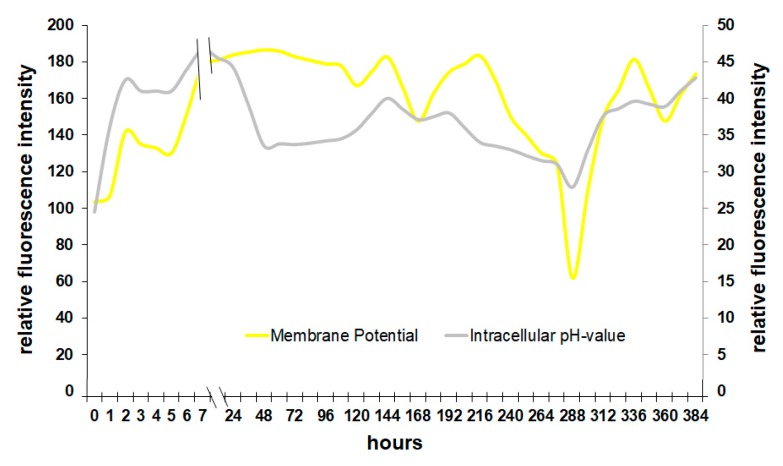
Correlation between membrane potential and intracellular pH during fermentation.

**Figure 6 microorganisms-08-00619-f006:**
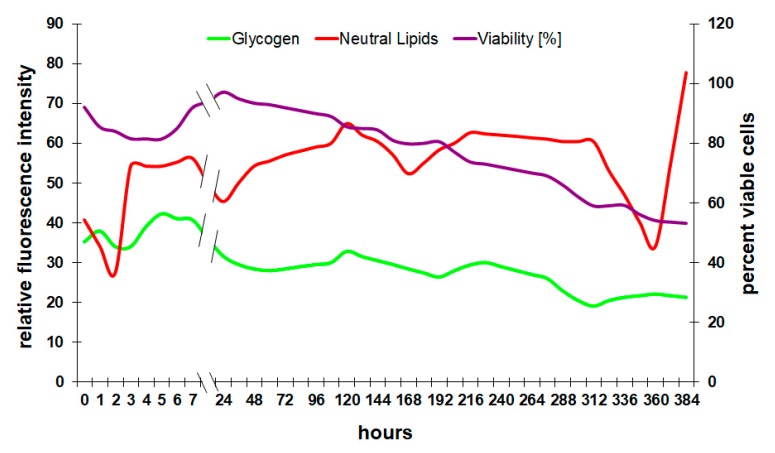
Course of reserve storage products during fermentation.

**Figure 7 microorganisms-08-00619-f007:**
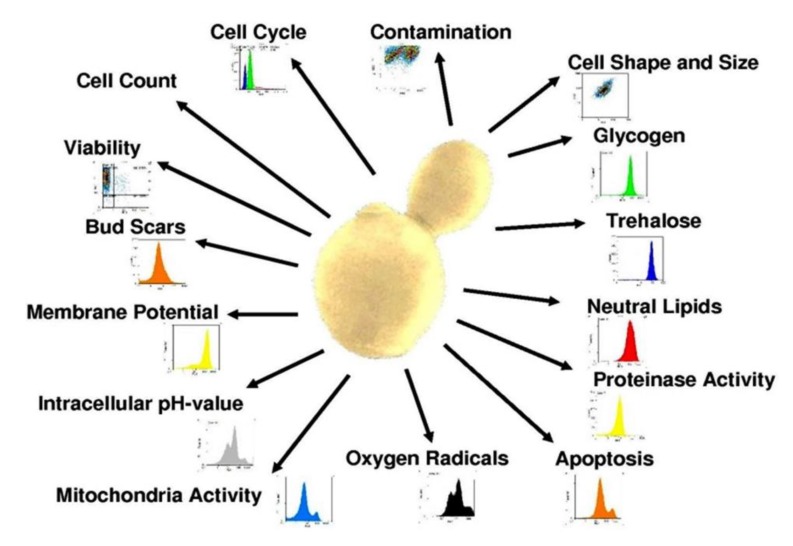
Yeast-related attributes that can currently be analyzed by flow cytometry.

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
