# Peer review of "Monitoring the Functionality and Stress Response of Yeast Cells Using Flow Cytometry"

_microorganisms, 2020, doi:10.3390/microorganisms8040619_

Round 1

Reviewer 1 Report

The research presented in this manuscript utilizes flow cytometry as an analytical instrument to understand ethanol fermentation performance by Saccharomyces cerevisiae. Fermentation performance metrics such as cell cycle, viability, glycogen content, DNA damage, and mitochondria activity were monitored by flow cytometry. Overall this research attempts to better understand how flow cytometry can be applied as an analytical tool that could be used in the fermentation industry, particularly by winemakers, as an extra option to improve ethanol fermentation performance. Although the manuscript is interesting and of interest to readers some revisions and comments need to be addressed by the author. 

  1. This reviewer recommends changing "Table 1" into a Figure. As it is right now it much more resembles a figure instead of a table. 
  2. The "Conclusion" sections needs to be rewritten. As it is currently written, the "Conclusion" section is too general. The "Conclusion" should include a few of the key results of the current study. 
  3. Figure 6 should be moved to a different part of the manuscript. This figure would be better placed in the "Results and Discussion" section, or utilized as a graphical abstract instead. 

Author Response

Thank you very much for the constructive review. I appreciate the suggestions. Please find my point-by-point response below.

  1. Table 1 was changed into Figure 1 and all the other Figures are changed accordingly.
  2. The conclusion was re-written and re-structured to make it more specific.
  3. The Figure was moved into the Results and Discussion section, including a paragraph that discusses the Figure.

Reviewer 2 Report

The paper is about how useful is a flow cytometry technique as a tool for monitoring yeast fermentation. The paper is interesting and gives the new utility of this technique.

However, some critical points need to be clarified or corrected before publication:

Section 2 Materials and Methods

I would like to know how many cells did the inoculum contain?  What volume did you take in each sample during the first 24 hours? What was the total volume of the fermentation? How many replicates of fermentation did you make?

Line 83 I don’t understand the usefulness of the observation “The other protocols use fresh cells, so dyeing can be done right after sampling and washing”.

Section 2.2 I think it would be interesting to give all reagents and chemicals in a table in the supplementary data.

Line 119 What temperature did you use in that incubation?

Line 179, 186, 189  I think it could need references

Figure 2  needs to improve because it is difficult to see. It would be interesting to know the values of the axes.

Line 246 I think it would be better resistant instead of resistible.

Line 263 but this increase in lipids at the end of the graph could also be explained by lipids released from the cell wall of dead cells.

Author Response

Thank you very much for the constructive review. I appreciate the comments. Please find my point-by-point responses below.

Materials and Methods: The requested information was added to the section.

Line 83: This detail distinguishes between cells that were fixed in ethanol and cells that can be dyed right after sampling and washing as viable cells. Protocols like cells cycle or glycogen content only work with fixed cells, so this detail is important.

Section 2.2: Thank you for the suggestion. However, I believe that the chemicals and reagents are sufficiently specified which will allow researchers to find them and repeat the experiments.

Line 119: The detail is now included in the description.

Lines 179, 186, 189: There are no references available that show that phenomenon. For this reason the Figure showing double-peaks was included in this manuscript. It visualizes the effect of inhomogeneous sampling.

Figure 2 (now Figure 3): The values on the axis are not relevant for this Figure because it only shows the progression of the two peaks, representing G1 and G2 cell during fermentation. The axis in the larger cell cycle analysis histogram in the bottom of the Figure is essentially the same since it is the same yeast culture and could be used as an axis reference.

Line 246: The text now says resistant instead of resistible.

Line 263: Please keep in mind that flow cytometry only analyzes cells and cell components since all medium residue is removed by washing the cells. Any lipids that are released into the medium by dead cells will not be included in the fluorescence signal.